# Small-RNA Sequencing Reveals Altered Skeletal Muscle microRNAs and snoRNAs Signatures in Weanling Male Offspring from Mouse Dams Fed a Low Protein Diet during Lactation

**DOI:** 10.3390/cells10051166

**Published:** 2021-05-11

**Authors:** Ioannis Kanakis, Moussira Alameddine, Leighton Folkes, Simon Moxon, Ioanna Myrtziou, Susan E. Ozanne, Mandy J. Peffers, Katarzyna Goljanek-Whysall, Aphrodite Vasilaki

**Affiliations:** 1Department of Musculoskeletal & Ageing Science, Institute of Life Course & Medical Sciences, Faculty of Health & Life Sciences, University of Liverpool, Liverpool L7 8TX, UK; M.Alameddine@liverpool.ac.uk (M.A.); peffs@liverpool.ac.uk (M.J.P.); kasia.whysall@nuigalway.ie (K.G.-W.); vasilaki@liverpool.ac.uk (A.V.); 2Chester Medical School, Faculty of Medicine and Life Sciences, University of Chester, Chester CH2 1BR, UK; i.myrtzioukanaki@chester.ac.uk; 3School of Biological Sciences, Faculty of Science, University of East Anglia, Norwich NR4 7TJ, UK; L.Folkes@uea.ac.uk (L.F.); s.moxon@uea.ac.uk (S.M.); 4Metabolic Research Laboratories, Wellcome-MRC Institute of Metabolic Science, University of Cambridge, Cambridge CB2 0QQ, UK; seo10@cam.ac.uk; 5Department of Physiology, School of Medicine and REMEDI, CMNHS, NUI Galway, Galway H91 TK33, Ireland

**Keywords:** maternal protein restriction, muscle development, offspring, microRNAs, sncRNAs, snoRNAs

## Abstract

Maternal diet during gestation and lactation affects the development of skeletal muscles in offspring and determines muscle health in later life. In this paper, we describe the association between maternal low protein diet-induced changes in offspring skeletal muscle and the differential expression (DE) of small non-coding RNAs (sncRNAs). We used a mouse model of maternal protein restriction, where dams were fed either a normal (N, 20%) or a low protein (L, 8%) diet during gestation and newborns were cross-fostered to N or L lactating dams, resulting in the generation of NN, NL and LN offspring groups. Total body and tibialis anterior (TA) weights were decreased in weanling NL male offspring but were not different in the LN group, as compared to NN. However, histological evaluation of TA muscle revealed reduced muscle fibre size in both groups at weaning. Small RNA-sequencing demonstrated DE of multiple miRs, snoRNAs and snRNAs. Bioinformatic analyses of miRs-15a, -34a, -122 and -199a, in combination with known myomiRs, confirmed their implication in key muscle-specific biological processes. This is the first comprehensive report for the DE of sncRNAs in nutrition-associated programming of skeletal muscle development, highlighting the need for further research to unravel the detailed molecular mechanisms.

## 1. Introduction

Intrauterine growth restriction and low birthweight are linked with a predisposition to adverse health consequences in adulthood, including insulin resistance, cardiovascular disease, hypertension and obesity [1]. Therefore, the concept of foetal and perinatal programming of later disease has been developed. Maternal diet during gestation and lactation plays a crucial role in embryonic and post-natal development of the offspring as it is the only source of nutrients through the placenta in pregnancy and provides all essential components for neonatal growth during lactation. Poor maternal nutrition during gestation and/or lactation is known to reduce growth and impair muscle development and stem cell activity [2,3,4], increase fat accretion [5] and alter metabolism [6] in the offspring. Offspring skeletal muscle development is susceptible to maternal nutrient restriction which also results in reduced offspring birth weight caused by decreases in foetal circulating amino acids [7,8]. Studies using ovine foetuses and offspring exposed to a poor nutritional environment in utero highlighted deficient muscle growth due to altered myofibre number and composition [9,10]. Further studies using mouse and rat models of maternal protein restriction suggested that this effect on muscle function may be long-lasting throughout the life-course [11,12,13]. There is also evidence from human studies of low muscle fibre score in elderly men born with low birth weight [14]. However, there is little understanding of the cellular and molecular mechanisms whereby environmental modulation in utero or in early postnatal life may lead to altered development of the musculoskeletal system [15,16].

The epigenetic regulation of skeletal muscle development and ageing has gained significant interest during the last decade [17,18]. Small non-coding RNAs (sncRNAs) are a class of epigenetic molecules including microRNAs (miRs), small nucleolar RNAs (snoRNAs) and small nuclear RNAs (snRNAs). These RNA molecules are fully functional, transcribed from DNA without processing for translation into proteins and, thus, act as crucial factors regulating gene expression. All three types of sncRNAs have been implicated in myogenesis and skeletal muscle development and regeneration [19,20]. miRs are non-coding RNAs (ncRNAs), 19–22 nucleotides (nt) in length, that regulate post-transcriptional gene expression and are linked to muscle development, disease and ageing [21,22]. Dietary interventions result in changes in miR expression in muscle [23], suggesting that the effects of diet on muscle may be mediated by miR-regulated changes in gene expression. miRs can simultaneously regulate many genes and signalling pathways, and therefore potentially physiological and pathological processes. It has been suggested that not all miRs are ubiquitously expressed, but many miRs are expressed in a tissue-specific manner [24,25]. There is a class of miRs in which their expression is directly associated with striated muscle, known as myomiRs [26], including miR-1, miR-133a, miR-133b, miR-206, miR-208b, miR-486 and miR-499 [27,28,29,30]. Although myomiRs have been extensively studied, other miRs may also be involved in regulating skeletal muscle development.

Ribosomes are considered as the cellular converters of the genetic codes, embedded in messenger RNAs, converting these codes into proteins [31]. It is known that ribosomal RNAs (rRNA) are post-transcriptionally modified via 2′O-ribose methylation or pseudouridylation [32]. These site-specific covalent modifications regulate the translational process. snoRNAs are mainly intron-derived, 50–250 nt long ncRNAs with high expression levels accumulating in the nucleolus. They are classified as C/D box snoRNAs (SNORDs), which are responsible for 2′-O-methylation of rRNAs, or H/ACA box snoRNAs (SNORAs), which guide pseudouridylation of nucleotides [33,34]. However, approximately half of human snoRNAs have no predictable rRNA targets, and numerous snoRNAs have been associated with diseases [35,36] that show no defects in rRNAs [37]. On the other hand, snRNAs have an average size of 150 nt, which localise in the nucleus and mainly control intron splicing [38]. snRNAs are present as ribonucleoprotein particles (snRNPs) forming a large complex (spliceosome) which mediate splicing by attachment to the unspliced primary RNA transcripts [39,40,41]. Despite the vital roles described, very little is known for the specific functions of snoRNAs and snRNAs in skeletal muscle physiology.

In this study, we aimed to identify miRs, snoRNAs and snRNAs that may regulate reduced skeletal muscle growth in the offspring due to maternal protein undernutrition during pregnancy or during lactation. We utilised a mouse model of maternal protein restriction and analysed small RNA-seq data derived from skeletal muscle of 21 days (21d) male offspring to determine sncRNAs that are differentially expressed at weaning. Furthermore, we validated some of the observed differentially expressed (DE) miRs by qPCR and performed an in silico bioinformatic analysis to unravel which signalling pathways and cellular functions may be affected by these miRs.

## 2. Materials and Methods

### 2.1. Animals

B6.Cg-Tg(Thy1-YFP)16Jrs/J mice, expressing yellow fluorescent protein (YFP) only in neuronal cells (Jackson Laboratory; stock number 003709), were used in this study. Mice were housed in individually vented cages maintained at 21 ± 2 °C on a 12 h light/dark cycle. All experimental protocols were performed according to the UK Animals (Scientific Procedures) Act 1986 regulations and obtained ethical approval from the University of Liverpool Animal Welfare Ethical Review Board (AWERB). Animal use followed the 3Rs guidelines. Mice were kept at the Biomedical Services Unit (BSU) of the University of Liverpool and monitored daily for any health and welfare issues.

### 2.2. Experimental Groups

Mice were fed ad libitum with solid food pellets containing normal protein proportion (N, 20% crude protein; Special Diet Services, UK; code 824226) or low protein proportion (L, 8% crude protein; Special Diet Services, UK; code 824248) with isocaloric value. Eight-week-old nulliparous female mice were fed on either N or L protein diet for two weeks before mating with age-matched males on N diet. The pregnant mice were kept on the same diet throughout gestation (19–21 days). Within 24 h after birth, male pups from N dams were cross-fostered with other N dams (NN group, *n* = 5) or L dams (NL group, *n* = 5) that were maintained under the same experimental conditions, but fed with L diet two weeks prior to mating and during gestation and lactation (21 days), whilst pups from L dams were transferred to N dams for lactation (LN, *n* = 5). Suckling pup numbers were kept the same for all dams during the 21d lactation period (*n* = 5 pups) to avoid confounding issues of litter size difference (Figure 1A). Male offspring were culled at the end of lactation by CO_2_ euthanasia and weighed and tibialis anterior (TA) skeletal muscles were immediately dissected and weighed.

### 2.3. Skeletal Muscle Histology

To characterise the skeletal muscle phenotype, TA muscles (*n* = 4/group) were mounted directly on a cork disk surrounded with Cryomatrix (Thermo Scientific^TM^, UK), immersed in liquid nitrogen-frozen isopentane (Sigma Aldrich, UK) for cryoprotection and stored at −80 °C until cryosectioning. TA muscles were placed at −20 °C for at least 30 min prior to cryosectioning. Transverse sections (10 μm) were cut using a Leica cryotome and collected on Superfrost glass slides (Thermo Scientific^TM^, UK). Sections were washed with PBS for 10 min before staining with 1:1000 dilution of rhodamine wheat germ agglutinin (WGA; 5 μg/mL; Vector Laboratories, UK) for 10 min, mounted in antifade medium (Vector Laboratories, UK) and were visualised with an Axio Scan.Z1 slide scanner (Zeiss, UK). Minimum Feret’s diameter (MFD) was measured using ImageJ software (U.S. National Institutes of Health, USA).

### 2.4. RNA Isolation, Library Preparation for Small RNA-Seq and Sequencing

TA skeletal muscles were ground using a mortar and pestle and liquid nitrogen. Total RNA was isolated and purified by the mirVanaTM kit (Thermo Scientific^TM^, UK) using the manufacturer’s protocols. Samples were processed with Clip-Clap^TM^ Acid Pyrophosphatase (Cambio, UK) prior to the library preparation to remove any 5′ cap structures, and size selected using a range 120–300 bp including adapters ensuring the unbiased identification of sncRNAs. Preparation of a small RNA-Seq library from submitted total RNA sample was performed using the NEBNext^®^ small RNA library preparation kit (New England Biolabs, UK). Small-RNA sequencing was conducted using the Illumina HiSeq 4000 (Illumina, San Diego, CA, USA) at 2 × 150-base pair (bp) paired-end sequencing, generating data from >280 M clusters per lane. Sequencing was performed at the Centre for Genomic Research, University of Liverpool (https://www.liverpool.ac.uk/genomic-research/, accessed on 9 May 2021).

### 2.5. Small RNA-Seq Data Processing

Mouse (mmu) mature miR sequences were downloaded from miRBase (v22.1) [42] in FASTA format. For the alignment of sRNA reads, the FASTA format miRBase sequence files were made non-redundant using java code and Uracil bases (U) were changed to thymine (T). Small RNA reads were converted from FASTQ to FASTA format and then processed to trim sequencing adaptors using the Perl script “Remove adaptors” [43], recognising the first 8 bases of the adapter sequence. The processed reads were then aligned to the processed mature miR sequences allowing zero mismatches using PatMaN [44] (parameters: -s -e 0 -g 0 and -s -e 1 -g 0). Custom java code was used to parse the alignment files and generate an aligned read count table across all samples. The DESeq2 [45] method within iDEP [46] (version 91) was used for normalisation of counts between samples and calling differentially expressed (DE) miRs using default settings.

To obtain sncRNA reference sequences, the *Mus musculus* (Genome Assembly GRCm38) annotation in gff3 format was downloaded from Ensemble [47] (release-96) from which all records representing a non-coding RNA gene with a snoRNA biotype were extracted and converted into bed format. The processed annotation file in bed format along with the Mus musculus genome top level assembly (Ensemble release-96, GRCm38) was used with the function “getfasta” within the bedtools software [48] (parameters: -s -name) to extract snoRNA sequences in FASTA format. The FASTA format snoRNA sequence file was then made non-redundant using java code. The processed sRNA reads were then aligned to the snoRNA sequences, aligned read count tables were generated and differential expression analysis was performed as described above. Java code used for sequencing processing along with count matrices are available online at https://github.com/lf-bioinformatics/sRNA-code, accessed on 9 May 2021.

Data were assessed using pairwise comparisons, while correlation heatmaps and principal component analysis (PCA) plots were visualised using the Clustvis web tool [49]. DE miRs were extracted by applying the threshold of false discovery rate (FDR) adjusted *p*-values < 0.05 (FDR < 0.05), generated using the Benjamini–Hochberg method [50] and an absolute value of log2 fold change of 1.0 (|log2FC| > 1, equating to a 2-fold change) in the first instance; thresholds of FDR < 0.001 and |log2FC| > 1.3 (equating to a 2.5-fold change) and FDR < 0.05 and |log2FC| > 1.3 were applied when narrowing the DE miR and snoRNAs ranges, respectively.

### 2.6. Quantitative Polymerase Chain Reaction (qPCR)

For miR expression analysis, total RNA was isolated and purified using the mirVanaTM kit (Thermo Fischer, UK), reverse transcription of total RNA containing miRs was performed with miScript II RT kit (Qiagen, UK). qPCR was performed on a RotorGene™ 6000 (Corbett Research) instrument in a 20 μL reaction mixture; qPCR conditions were: 95 °C for 30 s, 55 °C for 30 s and 72 °C for 30 s (40 cycles) using a hot start step of 95 °C for 15 s. Specific primers for miR-15a, -34a, -125b, -199 and -206 (Appendix A) were used for the qPCR utilising RNU6 as the reference gene [51,52]. The results were analysed using the modified delta CT method [53].

### 2.7. Bioinformatic Pathway Analysis of DE miRs and miR:Target Prediction

Potential biological connections of all DE miRs were identified using Ingenuity Pathway Analysis (IPA) (IPA, Qiagen Redwood City, CA, USA) “Core Analysis”. Putative miR-target gene prediction was performed by uploading DE miR data into the MicroRNA Target Filter module within IPA. This module combines experimentally validated databases, peer-reviewed publications and target gene predictions using TargetScan [54] to identify miR–mRNA interactions and creates biological networks describing functional associations. For each core analysis, “diseases and bio-functions” were queried. In addition, to predict gene targets of the selected miRs, the miRWalk on-line tool [55] was utilised by searching through four databases: miRWalk, TargetScan, miRDB [56] and MiRTarBase [57]. Cytoscape v3.7.2 [58] software was used to build the interaction networks between predicted targets and selected miRs as well as to determine the biological roles of the target mRNAs utilising Gene Ontology (GO) terms of biological process and molecular functions using the ClueGO plug-in [59]. KEGG pathway analysis [60] was performed to determine the implication of the predicted gene targets in biological pathways. *p* < 0.05 was considered to indicate a statistically significant result.

### 2.8. Statistical Analysis

All data were analysed with GraphPad Prism 6 software and expressed as the mean ± SD. Data sets were tested for Gaussian distribution with the D’Agostino–Pearson normality test. Comparisons between the NN and NL groups were performed by unpaired Student’s *t*-test or Mann–Whitney U test. Bioinformatic analyses statistics were performed with built-in packages in IPA and Cytoscape. The *p* values in GO analysis were calculated with Fisher’s Exact test corrected with Bonferroni post hoc test in Cytoscape. In all cases, *p* values less than 0.05 were considered statistically significant.

## 3. Results

### 3.1. Skeletal Muscle Phenotypic Characterisation of Male Offspring

Male offspring were collected at weaning (21d) and the size and total body weight were compared using the NN group as control (NN vs. NL and NN vs. LN). Our data indicate that mice born from dams maintained on a normal protein diet but fed postnatally by a foster dam maintained on a low diet (NL) showed significant reductions in body size (Figure 1B) and total body weight (Figure 1C) at 21 days compared to NN male mice. On the contrary, neonates born from dams on a low protein diet but were fed by dams on normal protein diet during lactation (LN) appeared to have equal mean size and total body weight to the NN group at weaning, as shown in Figure 1B,C. It is important to note that the total body weight of P0 (day of birth) neonates born from dams on low protein diet (L) was significantly lower than pups born from dams on normal protein diet (N) (Appendix A).

TA muscle weight index (expressing the net TA weight per total body weight of each pup) for NL offspring was lower compared to NN offspring, demonstrating a profound effect of maternal low protein consumption during lactation which was not observed in muscles from LN males (Figure 1D). To test if muscle size was associated with muscle fibre size, we compared the MFD in WGA stained cryosections from fibres in the midshaft of the TA muscle. We found that the MFD of TA muscle fibres from the NL and LN groups was significantly lower than the NN group (Figure 1E,F).

### 3.2. Differential Expression Analysis of miRs in TA Skeletal Muscle

Small RNA-seq was performed using total RNA isolated from TA muscles of male offspring at weaning. All RNA samples (500 ng) were of high purity and integrity with RNA Integrity Number (RIN) >7 as assessed on the Agilent 2100 Bioanalyzer system (Agilent Technologies Ltd., UK). An average of 14.18 million total reads for NN, 14.72 million for NL and 14.05 million for LN group was generated, of which 13.28%, 3.65% and 13.96% were mature miRs with mapped reads of 2.05 million, 0.51 million and 1.99 million, respectively. Normalised reads were used to estimate small RNA transcript expression of all samples, aiming to identify the most abundant sncRNAs. In total, 796 mature miRs were detected (Figure 2A). Only miRs with at least 10 raw reads in ≥60% of the samples in each group (≥3 samples/group) were considered measurable, and, after filtering with FDR ≤ 0.05 and |log2FC| > 1 (Fold Change), 179 miRs were extracted (Figure 2B). The DE of common miRs among the three groups, 145 in total, was statistically significant when NN and NL groups were compared, while no differences in miR expression levels were found between NN and LN groups. The only exception was miR-122 expression, which was slightly increased in LN group in comparison with NN (*p* = 0.046, Appendix A).

To reduce the high number of DE mature miRs in order to have a range of the most significant common miRs with the highest potency and a possible biological role, we applied an additional stricter filter of FDR < 0.001 and |log2FC| > 1.3 (>2.5-fold change), which resulted in 92 miRs (Appendix A). A PCA of NN, NL and LN samples using the DE of these 92 miRs indicated a significant difference of the NL group against the NN/LN groups (Figure 2C). The samples of the latter groups were overlapped in the PCA plot, indicating similarity in miR expression (Figure 2C). Finally, a heatmap revealed a clear difference in the distribution of these 92 common miRs between the NN/LN and NL groups of male mice (Figure 2D). Notably, two groups of samples were distinguishable: the DE of miRs from TA muscles of NL samples were assembled in one cluster, while NN and LN samples were presented as a combined cluster, indicating similar miR expression levels (Figure 2D).

### 3.3. Ingenuity Pathway Analysis for All DE miRs

To explore the potential biological associations of the 92 DE miRs in TA skeletal muscles from NL versus NN/LN male offspring, we undertook an IPA “Core Analysis”. Networks were generated based on information from Ingenuity Pathway Knowledge Database. Significant cellular functions determined by the DE miRs and predicted gene targets included “morphology of muscle cells” (*p* = 0.017) and “differentiation of muscle cells” (*p* = 0.008). The most interesting diseases associated with the constructed networks were “disarray of muscle cells” (*p* = 0.022) and “dystrophy of muscle” (*p* = 1.11 × 10^−6^) (Figure 3A). The top scoring network identified was “Organismal Injury and Abnormalities”, with *p* value ranging between 4.87 × 10^−2^ and 7.41 × 10^−6^ and involving 62 DE miRs. Finally, IPA revealed two statistically significant GOs associated with the DE miRs, namely “apoptosis of muscle cells” (*p* = 5.58 × 10^−8^) and “proliferation of muscle cells” (*p* = 1.36 × 10^−10^) (Figure 3B), as well as statistically significant GOs associated with predicted gene targets, e.g., “muscular hypertrophy” (*p* = 8.51 × 10^−8^) and “muscle contraction” (*p* = 1.5 × 10^−6^) (Figure 3C). A full list of statistically significant GOs associated with muscle is provided in Appendix A.

### 3.4. Selection and Validation of miRs from Small RNA-Seq

The next step was to select miRs which demonstrated a strong difference, in terms of both fold change DE and FDR, and among the NN, LN and NL groups of male mice which could have an impact of skeletal muscle development. We focused on four miRs: miR-15a, -34a, -122 and -199a. These specific miRs were selected based on our current work, the level of DE and following a literature search of the DE genes.

Among the 92 mature miRs with statistically significant DE, 42 were found with lower expression levels in NN/LN as compared to the NL group, including miR-122, while 50 were upregulated in NN/LN, including miR-15a, -34a and -199a (Figure 4A). Next, we tested if the selected miRs (miR-122, miR-15a, miR-34a and miR-199a) in combination with established and well-studied muscle-specific myomiRs [26], such as miR-1, -133a, -133b, -206, -208b, -486 and -499, known regulators of muscle cell proliferation and differentiation, can form a group of miRs able to profoundly distinguish the two groups of mice based on their expression levels in TA skeletal muscle. Indeed, a miR expression levels-generated heatmap, including these miRs, showed a clear DE of this group of miRs, highlighting the significant phenotypic difference in skeletal muscles affected by maternal protein restriction during lactation in NL males (Figure 4B). Finally, qPCR analysis was used to validate the miR-seq data. We found that miR-122 was expressed at higher levels in NL (Figure 4C), but not LN, a finding that contradicts the sequence data. miRs-15a, -199a and -34a expression levels were significantly lower in the NL mice, as shown in Figure 4D–F, respectively. These results are in agreement with miR-seq, while miR-206 expression levels were higher in NN than in NL and LN, which contradicts the sequence data (Figure 4G). In addition, miR-34a expression levels were found lower by qPCR in LN as compared to NN, a result which also opposes the miR-seq data (Figure 4B,F).

### 3.5. Selected miRs-15a, -34a, -122 and -199a Bioinformatic Pathway Analysis

Based on the observations and data described above, we performed a bioinformatic analysis to explore the possible molecular pathways and cellular functions that the predicted target genes of our selected mature miRs could be implicated in. Using a combination of strict parameters, i.e., seed length range 16–21 at the 3′-UTR site, binding *p* value of 1 and showing only the statistically significant mRNAs which occurred in all three miR databases (TargetScan, miRDB and MiRTarBase), we identified 70 predicted target genes in total for the four selected miRs (Figure 5A). GO analysis using the GlueGO plug-in of the Cytoscape software and including biological process, cellular functions and KEGG pathways revealed an interactive molecular network, as shown in Figure 5B. The six GO terms with the lower *p* values after Bonferroni corrections were: “cerebellum morphogenesis” (*p* = 0.0017), “embryonic pattern specification” (*p* = 0.0015), “regulation of striated muscle cell apoptotic process” (*p* = 4.2 × 10^−4^), “embryonic cranial skeleton morphogenesis” (*p* = 2.21 × 10^−4^), “positive regulation of cellular response to insulin stimulus” (*p* = 5.13 × 10^−5^) and “regulation of histone methylation” (*p* = 4.41 × 10^−5^) (Figure 5C).

Furthermore, to assess if our selected miRs interact with muscle-specific myomiRs, we used an in silico approach via Cytoscape, which revealed that this set of miRs appear to share some common gene targets forming an interrelated network (Appendix A). GO analysis showed that this combination of miRs (selected plus myomiRs) were involved in important developmental stages of the musculoskeletal system including motor neuron axon guidance and negative regulation of muscle tissue development (Appendix A).

### 3.6. Analysis of DE snoRNAs and snRNAs in TA Skeletal Muscle

We further analysed the small RNA-seq datasets to identify the DE snoRNA and snRNAs from TA skeletal muscles. Averages of 14.18 million total reads for NN, 14.72 million for NL and 14.05 million for LN group were generated, of which 12.77%, 14.95% and 10.88% were snoRNAs and snRNAs with mapped reads of 1.80 million, 2.22 million and 1.51 million, respectively. We detected 514 snoRNA and snRNAs in total, of which 95 were considered statistically significant, having FDR ≤ 0.05 and |log2FC| > 1.3 (Figure 6A). It is worth noticing that, of these 343 sncRNAs, 2 were identified as snRNAs (RNU3a and RNU73b), 39 H/ACA box snoRNAs (SNORA), 80 C/D box snoRNAs (SNORDs), 2 small Cajal body-specific RNAs (scaRNAs) and 220 uncharacterised (Figure 5B). After FDR and log2FC filter application, we used 2 snRNAs, 11 SNORAs, 26 SNORDs and 56 uncharacterised sncRNAs (Figure 6B and Appendix A). The analysis of normalised counts showed that Rnu3a was highly expressed in the TA muscles from NL group in comparison with either NN or LN. On the contrary, Rnu73a expression was significantly increased in NN and LN mice (Figure 6C). PCA analysis using DE SNORAs and SNORDs showed that NL samples were clustered tightly while NN and LN samples were grouped (Figure 6D). The heatmap representing DE revealed a clear difference in the distribution of SNORAs and SNORDs between the NN/LN and NL groups of male mice (Figure 6E).

## 4. Discussion

In this study, we report that maternal protein undernutrition during lactation (NL) causes a significant decrease in body mass, skeletal muscle weight and muscle fibre size in male offspring at weaning. This phenotype was accompanied by a very strong DE of numerous miRs as well as snRNAs and snoRNAs between NN and NL groups of mice. We also found that, among the plethora of altered miRs, there are four previously unreported DE miRs which strongly correlate with muscle-specific myomiRs and in silico bioinformatic analyses, implicating them in important developmental processes of the musculoskeletal system. Additionally, we demonstrated that mice born from dams fed a low protein diet during gestation but exposed to normal amounts of protein during lactation (LN) have a similar growth to the NN offspring with no apparent differences in body size, body weight or muscle weight at weaning. However, despite the lack of difference in weight, the TA myofibre size in these mice was reduced, which is similar to the reduction observed in the TA myofibres of the NL offspring.

There is growing evidence that the early-life nutritional environment, both in utero and in the early postnatal period, can have persistent long-lasting effects on later life health. Based on the developmental programming concept, this intergenerational effect can lead to high risk of obesity, type 2 diabetes and cardiovascular diseases [1,61]. Skeletal muscle is characterised by plasticity and is highly adaptive to nutritional changes by altering size, metabolic rates and functional properties. It is also affected in response to maternal nutritional imbalance by changes in offspring phenotype [10,62,63,64]. During foetal development, undernutrition can determine muscle fibre formation and growth in the offspring by suppression of muscle development associated gene expression [65]. Pregnancy and lactation are considered as critical periods, and it is evident that nutrient restriction before weaning may cause permanent changes in skeletal muscle [66]. Consistently, our results demonstrate that maternal low protein intake during lactation results in lower total body and skeletal muscle weight in NL male offspring. The result that pups at birth were found different in size and weight (depending on maternal protein diet during gestation) is in accordance with previous reports [67]. It appears that maternal diet during lactation can modify neonatal growth, since the LN group showed accelerated catch up growth and reached the same size and weight with the NN group at weaning, but not muscle fibre size. On the other hand, although P0 pups from mothers on normal protein diet are bigger, maternal low protein diet during lactation constrains their growth, resulting in smaller size of both total body and muscle fibres. During postnatal developmental stages, skeletal muscle growth is attained by muscle fibre size increase, and not muscle fibre number, mainly by satellite cells proliferation and myoblasts fusion [68], and muscle-specific net protein hypertrophic increase until 3 weeks of age [69]. Our results reveal that maternal low protein intake during pregnancy and/or lactation results in decreased size of myofibres, as assessed by fibre size measurements, consistent with other studies in rat [63]. This suggests that gestational as well as lactational protein feeding determines the skeletal muscle fibre size. This is crucial, since the myofibre number is still growing shortly after birth in rodents [70,71,72]. However, most studies have mainly focused on maternal high-fat diet [73] and cafeteria diets [74,75], but little is known about the mechanisms of low protein diet and how it affects muscle development in the offspring.

Although myogenesis is extensively studied, the focus has been limited to transcription factors and gene regulation. During the last decade, myomiRs have triggered the attention of many studies and next generation sequencing analyses have confirmed that myomiRs are abundant in muscle tissue [30]. MyomiRs are expressed in both cardiac and skeletal muscle with the exception of miR-206 [27], which is skeletal muscle-specific. Other studies have reported that some myomiRs are also expressed in other tissues at low levels, but it is accepted that their main roles are still in muscle [25]. For example, miR-486 is often referred to as “muscle-enriched” rather than “muscle-specific” because is expressed in other tissues as well [29]. In our study, we aimed to describe the miR signature that maternal low protein intake establishes in the skeletal muscle of male offspring. The skeletal muscle phenotypic differences between NN and NL groups was complemented with a profound change in miR signature, resulting in a clear distinguishable grouping of mice in PCA analysis and expression heatmaps. As expected, among the 92 DE miR that we found, after applying very strict statistical limits to ensure high accuracy of our results, myomiRs were included. Their expression levels during skeletal muscle development have been reported previously. miR-1 and miR-133a, for example, are upregulated by MyoD and MyoG during both human and mouse skeletal muscle differentiation in vitro [76,77] and in vivo [78,79] and also support myogenic differentiation [80,81]. Furthermore, miR-1/206/133 control embryonic myogenesis through regulation of BAF chromatin remodelling complexes [82]. Our findings are in agreement with these expression levels, i.e., miR-1 and -133a were significantly downregulated in the NL group compared with the NN group at weaning. It is also known that miR-1 and -206 suppress Pax7 expression and act as regulators of satellite cell proliferation and differentiation [81]. Postnatal muscle hypertrophy may also be regulated by satellite cell number per fibre at birth and their rate of proliferation as well as protein deposition (i.e., protein synthesis and degradation) [83]. Therefore, our future studies will focus on satellite cells which will be assessed in histological sections.

miR-206 was downregulated in NL mice and in LN mice too and validated by qPCR. It has been shown that miR-206 is highly expressed in newly formed muscle fibres, indicating that miR-206 may be involved in muscle regeneration and maturation [84]. Moreover, miR-206 directly targets Pax3, and inhibition with antagomirs results in delayed myogenic differentiation [85], but the miR-206 family is not essential for in vitro myogenesis, although it modulates the differentiation of skeletal myoblasts [86]. These results clearly demonstrate that the animal model used in the present study is valid and produces reliable sequencing data in accordance with the phenotypic outcome, since muscle size and miR expression are consistent with previous literature. Therefore, we proceeded to select and validate novel miRs that may have key functions in the nutrition-dependent developmental programming of skeletal muscle. It is reported that miR-15a has the highest expression level in skeletal muscle in comparison with other tissues [87] and is upregulated during human skeletal myoblast differentiation [88]. In addition, miR-34a targets thioredoxin reductase 2 (TXNRD2), which plays an important role in redox homeostasis of skeletal muscle [89] and is also involved in neuronal development [90]. Circulating miR-122 can enter muscle and adipose tissues in mice and reduce mRNA levels of genes involved in triglyceride synthesis and, thus, regulate energy balance [91]. Finally, miR-199a-3p expression is highly expressed in skeletal muscle and controls genes of the IGF1/Akt/mTOR signalling pathway to regulate C2C12 myoblasts differentiation [92].

It has also been previously reported that foetal exposure to maternal diabetes is associated with increased skeletal muscle expression of miR-15a, and that this may contribute to development of metabolic disease in later life [93]. Furthermore, miR-15a expression is elevated in powerlifters as compared to normal human controls [94] showing a positive correlation with increased muscle mass and strength. miR-34a was also downregulated in NL mice and LN mice, as shown by qPCR. miR-34a has been shown to induce senescence of endothelial progenitor cells by inhibiting SIRT1, which results in increased levels of acetylated FOXO1 [95] and, thus, inhibits skeletal muscle development. Other studies suggest that Sirt1 antisense (AS) long non-coding RNA (lncRNA) interacts with its mRNA to inhibit muscle formation by attenuating function of miR-34a [96]. Additionally, several reports describe that miR-34a plays important roles in neuronal development [89]. Specifically, miR-34a has been shown to regulate spinal morphology and neurite outgrowth and is associated with both morphological and electrophysiological changes in mouse models [97]. Ectopic expression of miR-34a has also been shown to modulate neural differentiation by increasing the percentage of post-mitotic neurons and neurite elongation of mouse neural stem cells, whereas antagomirs of miR-34a had the opposite effect, suggesting that miR-34a is required for proper neuronal differentiation [98]. Differences in the timing of changes in muscle innervation during development between human and in small mammals have been reported, thus in human the transition from poly- to mono-neuronal muscle innervation takes place between the 16 and 25 weeks of gestation, whereas in rodents these changes occur relatively later in the first two weeks after birth [99]. It is therefore possible that the reduction in miR-34a evident in both NL and LN groups is associated with dysregulation of muscle fibre innervation due to inadequate intake of protein during gestation or lactation and our future studies will focus on examining any specific changes occurring in muscle innervation in these mice phenotypically.

Although miR-122 is considered as a striated muscle-specific miR [25], its function has not been extensively explored. However, it is known to exhibit an inhibitory role of in the TGFβ/Smad signalling pathway in skeletal muscle fibrosis after contusion [100]. Finally, miR-199a regulates myogenic differentiation by acting downstream of Srf, which targets multiple factors within the Wnt signalling pathway [79]. Based on this literature, we selected miRs-15a, -34a, -122 and -199a and examined the DE of miR-seq by qPCR. Our bioinformatic approaches revealed that our validated miRs are predicted to target a variety of genes that regulate different developmental processes including striated muscle cell apoptosis and histone methylation, which is associated with epigenetic regulation of embryonic myogenesis [101]. Importantly, these miRs share a variety of common predicted gene targets that are implicated in skeletal muscle physiology with myomiRs. We, therefore, conclude that the four selected miRs can serve as a complement of known myomiRs and form a promising set to monitor skeletal muscle development, but further studies need to confirm this idea. Considering that tissue-specific miR expression can be affected by the nutritional status and, in this case, the maternal low protein consumption particularly during lactation, we can conclude that the DE of these miRs might be the cause of reduced myofibre size in NL mice. The present study also highlights the importance of a maternal balanced diet especially during lactation as this has a direct impact on offspring health. A limitation of our work is that we have not examined the role of our selected miRs in vitro, e.g., by using C2C12 cells to evaluate fusion and myofibre formation, or in vivo by direct injections of miRs or antagomiRs and assessment of skeletal muscle integrity throughout life course. However, these approaches are beyond the scope of our present study and we intend to include them in our future work.

The increase in skeletal muscle mass after birth occurs primarily through muscle fibre hypertrophy given that fibre number is established during early postnatal development [102]. The amount of protein synthesised by a cell is dependent on translational capacity and efficiency. Ribosome biogenesis is as an important regulator of skeletal muscle growth and maintenance by changing the cellular translational capacity. There is a role for ribosome biogenesis in skeletal muscle growth during postnatal development [103]. In mammals, the ribosome is assembled from ribosomal RNAs (rRNAs), together with at least 80 different protein subunits [31]. Within the ribosome, 18S rRNA guides the decoding of the mRNA, whereas 28S rRNA forms the core of the peptidyltransferase centre that polymerises the amino acid sequence encoded by the mRNA into functional proteins. Maturation post-transcriptionally of rRNAs is part of the biosynthesis of ribosomes [104]. 47S rRNA ribonucleolytic processing into mature 18S, 5.8S and 28S rRNAs is rate-limiting for ribosome biogenesis [105]. We identified an increase in U3 in NL compared to NN. Whilst a network of snoRNAs is involved in key ribosome biogenesis processes [106], U3 is a factor in the generation of 18S rRNA [106]. This snoRNA is a highly abundant and evolutionarily conserved box C/D-class snoRNA, guiding the endoribonucleolytic processing of the 5′ external transcribed spacer (ETS) of the 47S pre-rRNA by base complementarity-guided pre-rRNA substrate recognition [107]. It has a key role in the maturation of 18S rRNA [108,109]. In cartilage, we demonstrated the global impact of reduced U3 expression on protein translational processes and inflammatory pathways and found that altering U3 expression has a direct effect on rRNA expression and translational capacity of chondrocytes. We showed that the extracellular environment is capable of controlling cellular U3 levels, thereby tuning the cell’s capacity to generate mature rRNA species [110]. The increase in U3 in NL may be a compensatory method to protein restriction.

Although miRs have been extensively studied, very limited evidence refers to the role of non-miR sncRNAs in muscle development [111]. Several SNORDs, such as SNORD-116, -48, -84, -95 and -97, showed DE in skeletal muscle of Amyotrophic Lateral Sclerosis (ALS) patients as compared to healthy controls [112]. RNA-seq analysis using serum and muscle samples of Duchenne Muscular Dystrophy (DMD) patients also revealed altered snoRNA and snRNA levels, suggesting their use as potential biomarkers for DMD [113]. Here, we report that low protein intake during lactation in NL male mice results in DE of multiple snoRNAs of both C/D and H/ACA box subtypes with canonical roles, for the first time in the literature. Two snoRNAs (SNORD33 and SNORD35) were reduced in NL. Whilst these snoRNAs have canonical roles [114], they additionally have non-canonical roles as regulators of metabolic and oxidative stress pathways in mammalian cells as they were induced in vivo in response the generalised oxidative stress [115]. Loss of their host gene Rpl13a altered mitochondrial metabolism and lowered reactive oxygen species (ROS) production, but the snoRNAs are themselves regulated by reactive oxygen species and oxidative stress, with dynamic accumulation of snoRNAs in the cytoplasm in response to oxidative stress [116]. In human knee cartilage, we identified changes in these “oxi-snoRNAs” in ageing and osteoarthritis, and this was a cross species effect. Our in vitro experiments demonstrated that they respond to their environment and oxidative stress stimuli and using physiological levels of ROS generated using fibronectin fragments increased the expression these snoRNAs [117]. In skeletal muscle, suboptimal maternal nutrition followed by accelerated postnatal growth has been shown to induce an accelerated ageing phenotype and oxidative stress in adult offspring male rats. Specifically, Tarry-Adkins et al. demonstrated that rats born to dams fed on low protein chow but suckled postnatally by a dam maintained on a normal diet until weaning and maintained on a normal diet until 12 months of age have accelerated telomere shortening and increased DNA damage, which was associated with a strong oxidative stress phenotype, a compensatory increased antioxidant defence enzyme activity and inflammation [118]. The authors also demonstrated that mitochondrial dysfunction was evident in the muscle of the offspring by a reduction in citrate synthase activity (a marker of intact mitochondria) and increased Complex I, linked Complex II III and Complex IV electron transport chain (ETC) activity, suggesting that muscle mitochondria might have to compensate for fewer mitochondria by increasing the activity of ETC complexes to generate sufficient ATP, which in turn produces more ROS. Although we did not examine these changes directly in this study, we speculate that these ROS-related mechanistic changes occur as a consequence of suboptimal in utero and early environments and are affected directly by oxi-miRs and/or oxi-snoRNAs, and this in turn leads to the development of ROS induced age-related diseases including sarcopenia [118].

## 5. Conclusions

We here report that small RNA-seq reveals numerous sncRNAs which are correlated with reduced skeletal muscle size of 21d male offspring that were lactated from mouse dams on low protein diet. We further describe four differentially expressed miRs, miRs-15a, -34a, -122 and -199a, that may be implicated in this process. These selected miRs could be combined with established myomiRs composing an interactive biomolecular network that may regulate this effect, as the bioinformatic approach revealed. Finally, to our knowledge, this is the first study exploring the role of snoRNAs in muscle development under restricted diet conditions. Although we do not provide a mechanistic explanation or a functional outcome of these alterations, our study suggests novel sncRNA signatures and identify potential candidates for innovative directions in muscle development. Future studies will facilitate uncovering the detailed molecular mechanisms.

## Figures and Tables

**Figure 1 cells-10-01166-f001:**
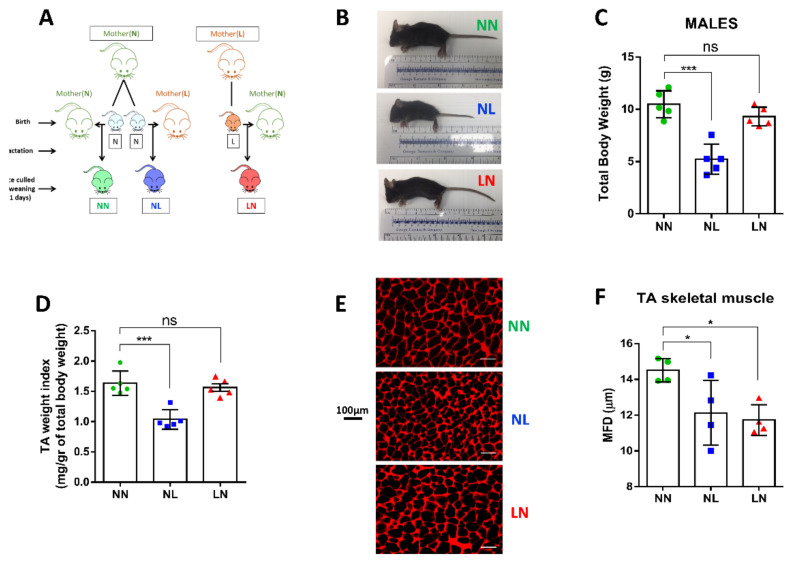
Experimental design and skeletal muscle morphology. Illustrative description of the experimental design of the study (**A**). Male 21d offspring lactated by dams on low protein diet (NL) were smaller in size (**B**) and had lower total body (**C**) and TA muscle/body weight index (**D**) compared to the NN group, while offspring lactated by dams on normal protein diet (LN) did not differ from NN control group. Histological assessment revealed that TA myofibre size was decreased in both NL and LN groups, as shown in representative images (**E**) and measurements of muscle fibre minimum Feret’s diameter (MFD) (**F**). All data are presented as mean ± SD. * *p* < 0.05, *** *p* < 0.001, ns, non significant. Scale bar: 100 μm.

**Figure 2 cells-10-01166-f002:**
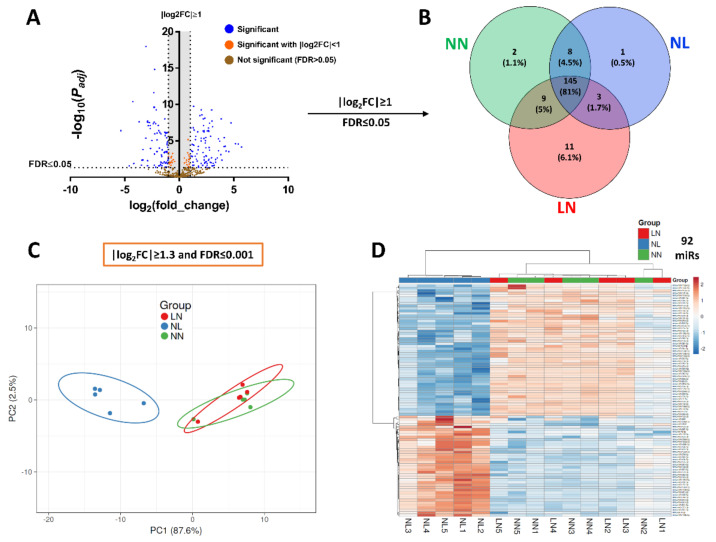
Differential expression of miRs. Small RNA-seq data analysis (*n* = 5/group) showed differential expression (DE) of numerous miRs, illustrated by Volcano plot (**A**). After filtering with FDR ≤ 0.05 and |log2FC| ≥ 1, there were 179 remaining DE miRs, where 145 were common among NN, NL and LN groups (**B**). Application of additional and stricter filters, FDR ≤ 0.001 and |log2FC| ≥ 1.3, narrowed the number of common DE miRs to 92 (see Appendix A); PCA analysis indicated a profound difference between NN and NL groups (**C**), whilst NN and LN were mapped together. These differences were also reflected in expression heatmap of these 92 miRs (**D**). The colour of each entry is determined by the number of reads, ranging from red (positive values) to blue (negative values).

**Figure 3 cells-10-01166-f003:**
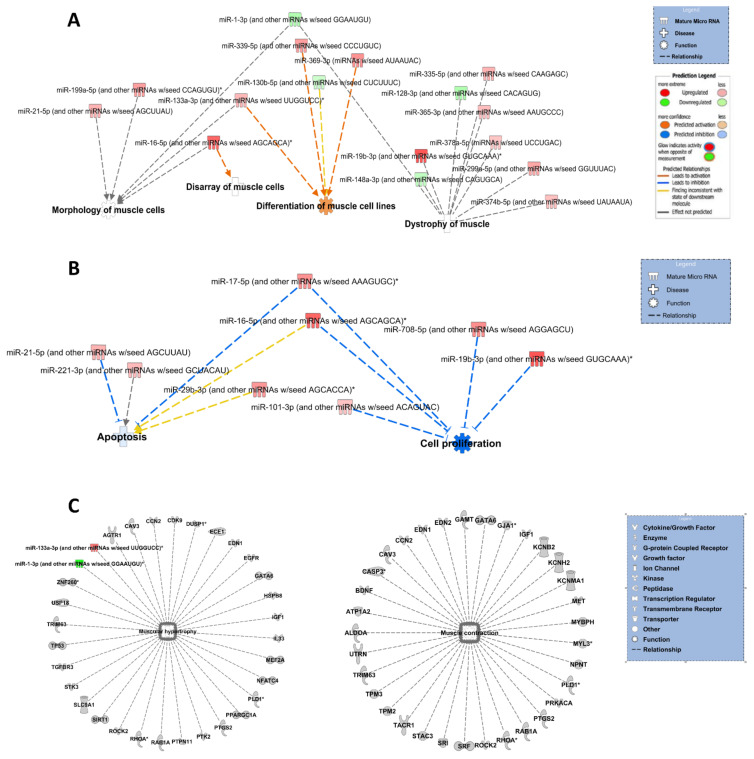
Bioinformatic analyses of differentially expressed miRs. Ingenuity Pathway Analysis representation of functions using DE miRs between NN/LN and NL male groups. Differentiation, morphology and disarray of muscle cells were identified by IPA as the predominant cellular functions and dystrophy of muscle as the most relevant disease which were correlated with the 92 differentially expressed miRs (**A**). Top GO terms identified in miR IPA core analysis were apoptosis and cell proliferation which relate to specific miRs (**B**). Muscle hypertrophy and contraction were the top GO terms associated with predicted gene targets of DE miRs (**C**). Figures are graphical representations of miRs identified in our data in their respective networks. Red nodes correspond to upregulated gene expression in NN/LN, and green nodes downregulation. Intensity of colour is related to higher fold-change. Legends to the main features in the networks are shown. The function colour is dependent on whether it is predicted to be activated or inhibited.

**Figure 4 cells-10-01166-f004:**
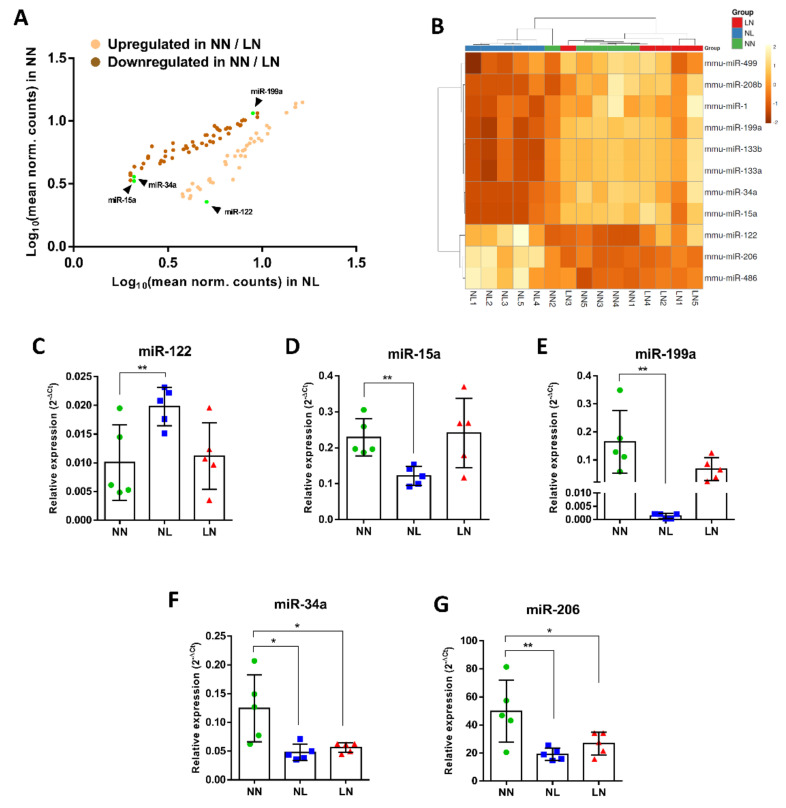
Validation of selected differentially expressed miRs by qPCR. Among the 92 miRs with DE between NN/LN and NL as determined by small RNA-seq, miR-122 expression was higher in NL, whereas miR-15a, -34a and -199a were found upregulated in NN and LN (**A**). The combination of these four miRs with known myomiRs revealed an interactive set of skeletal muscle-associated miRs in which the degree of DE between NN/LN and NL is very strong, as illustrated by the expression heatmap (**B**). The colour of each entry is determined by the number of reads, ranging from yellow (negative values) to brown (positive values). To confirm the expression levels, validation by qPCR showed increased expression in NL, but not in LN, for miR-122 (**C**). The results were similar between small RNA-seq and qPCR for miR-15a (**D**) and miR-199a (**E**), while for miR-34a (**F**) and miR-206 (**G**) were different between the two detection systems. * *p* < 0.05, ** *p* < 0.01.

**Figure 5 cells-10-01166-f005:**
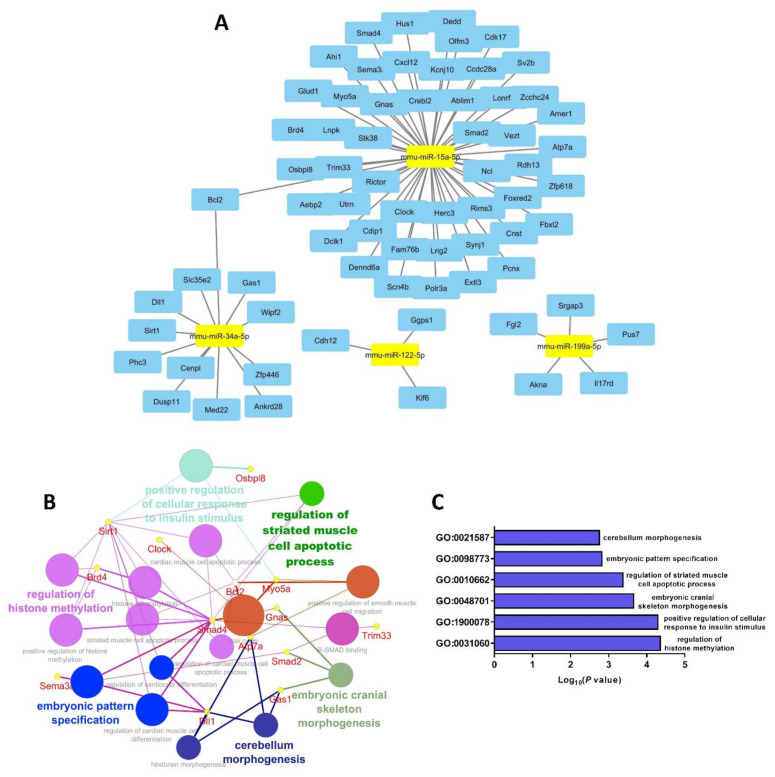
Bioinformatic network of selected miRs predicted gene targets. Target analysis of miR-15a, -34a, -199a and -122 by Cytoscape showed numerous (70) predicted gene targets (**A**). KEGG pathway analysis of the 70 predicted genes indicated that they could interact with each other forming a molecular network that can regulate significant as well as diverse developmental processes. Small circles represent a range of *p* value of 1.5–1.7 × 10^−3^, moderate circles 2.2–4.2 × 10^−5^ and large circles 4.4–5.1 × 10^−5^ after Bonferroni correction (**B**). The most significant GO showing statistical significance were cerebellum morphogenesis, embryonic pattern specification, regulation of striated muscle cell apoptotic process, embryonic cranial skeleton morphogenesis, positive regulation of cellular response to insulin stimulus and regulation of histone methylation (**C**).

**Figure 6 cells-10-01166-f006:**
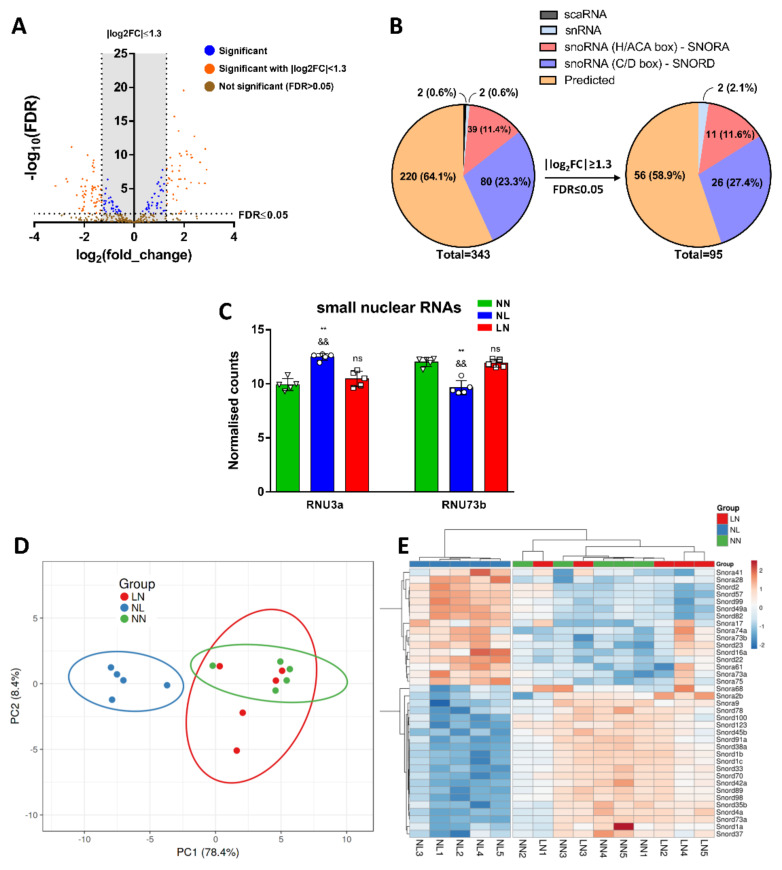
Bioinformatic analyses of differentially expressed sn/snoRNAs. Small RNA-seq data analysis revealed DE of multiple snRNAs and snoRNAs, as shown in the Volcano plot (**A**). Filters application of FDR ≤ 0.05 and |log2FC| ≥ 1.3 indicated the residual 95 DE sncRNAs, of which 56 were uncharacterised (**B**). The expression levels of snRNAs showed statistically significant upregulation of RNU3a and downregulation of RNU73b in NL group in comparison with NN and LN groups (**C**). PCA analysis demonstrated a profound difference between NN and NL groups (**D**), whilst NN and LN were clustered together. DE expression levels of SNORAs and SNORDs were plotted as heatmap suggesting distinct clusters of NN/LN and NL samples (**E**). The colour of each entry is determined by the number of reads, ranging from blue (negative values) to red (positive values). ** *p* < 0.01 as compared to NN, ^&&^ *p* < 0.01 as compared to LN; ns, not significant.

## Data Availability

Raw sequencing data for this study are stored at the Sequence Read Archive (SRA) using the BioProject accession: PRJNA668275.

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
