# Peer review of "Small-RNA Sequencing Reveals Altered Skeletal Muscle microRNAs and snoRNAs Signatures in Weanling Male Offspring from Mouse Dams Fed a Low Protein Diet during Lactation"

_cells, 2021, doi:10.3390/cells10051166_

Round 1
Reviewer 1 Report
This is a very interesting manuscript on the mechanisms of fetal programming of later disease, under conditions of maternal protein restriction. Authors studied expression of different groups of sncRNAs, not only miRNAs but also snRNAs and snoRNAs which should be considered as a bonus since most works only analyse miRNAs. The bioinformatic analysis is quite complete and authors also validated hits by qPCR.
I only have a substantial concern on this work, regarding the detected mRNA targets of the selected miRNAs. Since no data is presented, but only the KEGG/GO pathway analysis, it is difficult to ascertain the mechanisms of miRNA/mRNA involved, e.g. do these include any myogenic master gene? Any transcription or regulatory factor? I think that it would be very interesting to cite the 70 predicted targets for the selected miRNAs. Furthermore, I would also acknowledge that the discussion would be centered on the actual miRNA/mRNA interactions from the data generated in this work and not on “general” interactions, i.e. were MyoD/MyoG or Pax3 among the predicted targets of the myomiRs detected?
Minor concerns
-Why do authors use mice expressing yellow fluorescent protein (YFP) in neuronal cells?
-How were ncsRNAs size selected prior to library construction?
-There are a number of weird typographical mistakes in the References section, Refs. 28, 53 and 108
Author Response
Reviewer #1:
This is a very interesting manuscript on the mechanisms of fetal programming of later disease, under conditions of maternal protein restriction. Authors studied expression of different groups of sncRNAs, not only miRNAs but also snRNAs and snoRNAs which should be considered as a bonus since most works only analyse miRNAs. The bioinformatic analysis is quite complete and authors also validated hits by qPCR.
I only have a substantial concern on this work, regarding the detected mRNA targets of the selected miRNAs. Since no data is presented, but only the KEGG/GO pathway analysis, it is difficult to ascertain the mechanisms of miRNA/mRNA involved, e.g. do these include any myogenic master gene? Any transcription or regulatory factor?
We thank the reviewer for this interesting comment. Indeed, as a holistic approach, our ultimate goal for this project is to unravel the molecular mechanisms that regulate maternal protein restriction-induced skeletal muscle defects in the offspring. To achieve that, we intend to perform RNA-seq and assess the DE expression of the target genes which, potentially, relate to the sncRNAs described in this manuscript. However, the goal of the current manuscript is to report the phenotypic observations and initially identify the sncRNAs that may be implicated. These next steps are included in our future plans and are outside the scope of this manuscript.
I think that it would be very interesting to cite the 70 predicted targets for the selected miRNAs.
Thanks for the comment. These genes are presented in Fig 5A.
Furthermore, I would also acknowledge that the discussion would be centered on the actual miRNA/mRNA interactions from the data generated in this work and not on “general” interactions, i.e. were MyoD/MyoG or Pax3 among the predicted targets of the myomiRs detected?
Thanks for this comment. We prefer to make a more “general” discussion at this stage having these results in our hands. Muscle-related genes like Myo5a and Sirt1 seem to be predicted targets for our selected miRs, but this to be further investigated. In addition, other important muscle regulators, e.g. Igf1 and Pax7 (Table S3) are listed as predicted targets of the set of DE miRs we report here. As stated above, we aim to describe the DE of the sncRNA transcriptome in this paper and then deepen into specific pathways in future works.
Minor concerns
-Why do authors use mice expressing yellow fluorescent protein (YFP) in neuronal cells?
We used these transgenic mice as part of a BBSRC-funded project exploring the effects of maternal protein diet on the skeletal muscles and neuromuscular junctions (NMJ) in the offspring. These YFP transgenic mice is an excellent tool to visualise NMJs both in vivo and ex vivo. The 3Rs guidelines were followed (added in Lines 107-108).
-How were ncsRNAs size selected prior to library construction?
According to the Centre of Genomic Research (CGR, University of Liverpool), where the small RNA-seq was performed, the size for sncRNAs was selected at a range of 120-300bp. Given that the adapter was 120bp, the peak at 120-150bp (adapter ligated) corresponds to miRs population. The range is referred in Line 142 in the original manuscript.
-There are a number of weird typographical mistakes in the References section, Refs. 28, 53 and 108
These are now corrected.
Reviewer 2 Report
Kanakis et al present their work characterising the small non-coding RNAs in skeletal muscle on cohorts of differing protein-level diets during lactation. Small RNA-Seq and bioinformatics analyses identified significantly differentially expressed miRNAs and snoRNAs.
This paper makes a worthy contribution to the study of the often ignored, but crucially important small RNAs.
Major comments
Do any of the identified DE miRNAs originate from known clusters of miRNAs or are they randomly distributed throughout the genome? Plotting miRNA location by chromosome position would reveal this (e.g. with karyoplotR). miRNAs from the same clusters could indicate shared regulation. The same would also be of interest for the snoRNAs.
Confirmation of expression differences for a selection of the predicted miRNA target genes would go some way to validating these predictions. This could either be via qPCR, or if available, for the same feeding model, through publicly available mRNA-Seq.
It is good to see the authors have put their code on GitHub to help reproducible science. I’d also like to see the count matrix uploaded to either GitHub or GEO with the raw data (I was unable to access the raw data). This would remove the computational burden for researchers without sufficient resources or technical ability to align and QC the data.
Minor comments
- Methods: details of the trimming were not provided. Was a tool like trimmomatic or trim_galore used?
- Figure 2D. The miRNA gene names are too small to be read
- Figure4B: What are the unites for the scale on the heatmap?
- Figure 5B: Add to the legend the significance of the different sized circles. The numbers of genes per term are given in the supplemental information, but I think it is appropriate to also include this in the figures.
- Figure 6B. The grey shades are difficult to distinguish from the legend to the pie charts
Author Response
Reviewer #2:
Kanakis et al present their work characterising the small non-coding RNAs in skeletal muscle on cohorts of differing protein-level diets during lactation. Small RNA-Seq and bioinformatics analyses identified significantly differentially expressed miRNAs and snoRNAs.
This paper makes a worthy contribution to the study of the often ignored, but crucially important small RNAs.
Major comments
Do any of the identified DE miRNAs originate from known clusters of miRNAs or are they randomly distributed throughout the genome? Plotting miRNA location by chromosome position would reveal this (e.g. with karyoplotR). miRNAs from the same clusters could indicate shared regulation. The same would also be of interest for the snoRNAs.
We thank the reviewer for this comment. We assume that, due to the large number of statistically significant DE miRs identified in this work, mapping of these miRs according to their chromosomal loci will most likely reveal miR clusters with shared regulatory roles and will indlude this suggestion in our future work. However, in this manuscript, we intend to report these numerous miRs and explore their function(s) in later work(s). We have already started in vivo experiments with selected injectable miRs to explore if maternal protein diet-induced skeletal muscle dysregulation in the offspring can be reversed and this comment is of great value, however this work is outside the scope of this manuscript.
Confirmation of expression differences for a selection of the predicted miRNA target genes would go some way to validating these predictions. This could either be via qPCR, or if available, for the same feeding model, through publicly available mRNA-Seq.
Thanks for this interesting comment. Our goal for this large project is to reveal the molecular mechanisms that encompass skeletal muscle deficiency in the offspring due to maternal protein malnutrition. Naturally, RNA-seq is one of our next steps to attain this purpose at the transcriptional level and evaluate the DE expression of the target genes of these miRs. We believe the qPCR validation would lead to a bias in the current model through selection of genes known to be important in myogenesis, whilst we are undertaking an unbiased approach which may lead to discovery of novel regulators of muscle development.
It is good to see the authors have put their code on GitHub to help reproducible science. I’d also like to see the count matrix uploaded to either GitHub or GEO with the raw data (I was unable to access the raw data). This would remove the computational burden for researchers without sufficient resources or technical ability to align and QC the data.
We thank the reviewer for this comment. We have changed the release setting on SRA and the raw data is now publicly available and can be accessed. As suggested we have uploaded the count matrices to GitHub and edited the manuscript accordingly (Line 172).
Minor comments
- Methods: details of the trimming were not provided. Was a tool like trimmomatic or trim_galore used?
We thank the reviewer for the question. The raw FASTQ files were trimmed to remove Illumina adapters using the published Perl script ‘Remove adapters’ available in the supplementary material of Fowler et al.
[43] Fowler, E.K.; Bradley, T.; Moxon, S.; Chapman, T. Divergence in Transcriptional and Regulatory Responses to Mating in Male and Female Fruitflies. Sci Rep 2019, 9, 1610, doi.org/10.1038/s41598-019-51141-9
We have revised the text accordingly as well as added the reference (Lines 155-156).
- Figure 2D. The miRNA gene names are too small to be read
Due to the large number of DE miRs, we could not enlarge more the Figure. This is the reason, a detailed list for all 92 miRs is provided as Table S2 and the corresponding Table S5 for snoRNAs.
- Figure4B: What are the unites for the scale on the heatmap?
Many thanks for this observation. A comment for the scale colours is now added in each of the legends to Figure 2D (Lines 278-279), Figure 4B (Lines 336-337) and Figure 6E (Lines 397-398).
- Figure 5B: Add to the legend the significance of the different sized circles. The numbers of genes per term are given in the supplemental information, but I think it is appropriate to also include this in the figures.
The range of p values for the different sized circles is now added to the legend in Fig5 (Lines 359-361).
- Figure 6B. The grey shades are difficult to distinguish from the legend to the pie charts
This is now corrected.